**Subject Area:**
cellular biology/genomics/genetics/molecular biology

neuroblastoma, circulating biomarkers, cell-free DNA, ALK, *MYCN*

**Author for correspondence:**
Ricky M. Trigg
e-mail: rt473@cam.ac.uk

# Opportunities and challenges of circulating biomarkers in neuroblastoma

Ricky M. Trigg[1], Jacqui A. Shaw[2,†] and Suzanne D. Turner[1,†]

[1]Division of Cellular and Molecular Pathology, Department of Pathology, University of Cambridge, Cambridge CB2 0QQ, UK
[2]Leicester Cancer Research Centre, College of Life Sciences, University of Leicester, Leicester LE2 7LX, UK

RMT, 0000-0001-9329-9344; JAS, 0000-0003-4227-503X; SDT, 0000-0002-8439-4507

Molecular analysis of nucleic acid and protein biomarkers is becoming increasingly common in paediatric oncology for diagnosis, risk stratification and molecularly targeted therapeutics. However, many current and emerging biomarkers are based on analysis of tumour tissue, which is obtained through invasive surgical procedures and in some cases may not be accessible. Over the past decade, there has been growing interest in the utility of circulating biomarkers such as cell-free nucleic acids, circulating tumour cells and extracellular vesicles as a so-called liquid biopsy of cancer. Here, we review the potential of emerging circulating biomarkers in the management of neuroblastoma and highlight challenges to their implementation in the clinic.

## 1. Introduction to neuroblastoma

Neuroblastoma (NB) is the most common and deadly solid extracranial malignancy in children, deriving from precursor sympathoadrenal cells of the neural crest [1,2]. NB is often termed a 'clinical enigma', owing to its heterogeneous clinical behaviour ranging from spontaneous regression to treatment resistance, metastasis and death [3,4]. Therefore, in addition to the post-surgical International Neuroblastoma Staging System (INSS; table 1), patients are classified as very low (approx. 28%), low (approx. 27%), intermediate (approx. 9%) or high risk (approx. 36%) based on the likelihood of disease progression and relapse [5–8]. While low-risk disease is usually diagnosed in children younger than 18 months and may spontaneously regress without treatment, high-risk disease generally occurs in children older than 18 months and around half of patients relapse [9]. Despite intensive multi-modal treatment, 5 year survival among high-risk patients remains at 40–50% [10].

Genomic amplification of *MYCN* is reported in around 25% of NB tumours (approx. 40% among high-risk patients) and is generally accepted as the strongest predictor of poor prognosis and rapid tumour progression [11,12]. Other poor prognostic features include chromosome arm-level alterations, namely deletions of 1p (30%) and 11q (45%) and unbalanced gain of 17q (60%), all of which are associated with diploid or near-tetraploid karyotypes [13–16]. In addition, amplification of *ALK*, encoding the anaplastic lymphoma kinase (ALK) receptor tyrosine kinase, is observed in 1–2% of cases and is often co-amplified with *MYCN* [17–19]. Recently, massive genomic rearrangement, known as chromothripsis, has been observed in 18% of advanced stage tumours; thus, NB could be considered a predominantly copy number-driven cancer [20,21]. Somatic mutations are less common and include point mutations of *ALK* (8–10%) as well as point mutations and small, in-frame deletions of alpha thalassaemia/mental retardation syndrome X-linked (*ATRX*), encoding a SWI/SNF family chromatin remodelling protein, that occur at a frequency correlating with age at diagnosis [20,22–25]. *ATRX* alterations are associated with poor prognosis [24]. Recent genome-wide sequencing analyses in large

†These authors contributed equally to this work.

**Table 1.** International Neuroblastoma Staging System (INSS).

royalsocietypublishing.org/journal/rsob    Open Biol. 9: 190056

| stage | definition |
|---|---|
| 1 | Localized tumour with complete gross excision, with or without microscopic residual disease; representative ipsilateral lymph nodes negative for tumour microscopically (nodes attached to and removed with the primary tumour may be positive). |
| 2A | Localized tumour with incomplete gross excision; representative ipsilateral non-adherent lymph nodes negative for tumour microscopically. |
| 2B | Localized tumour with or without complete gross excision, with ipsilateral non-adherent lymph nodes positive for tumour. Enlarged contralateral lymph nodes must be negative microscopically. |
| 3 | Unresectable unilateral tumour infiltrating across the midline,[a] with or without regional lymph node involvement; or localized unilateral tumour with contralateral regional lymph node involvement; or midline tumour with bilateral extension by infiltration (unresectable) or by lymph node involvement. |
| 4 | Any primary tumour with dissemination to distant lymph nodes, bone, bone marrow, liver, skin and/or other organs (except as defined for stage 4S). |
| 4S | Localized primary tumour (as defined for stage 1, 2A or 2B), with dissemination limited to skin, liver and/or bone marrow[b] (limited to infants < 1 year of age). |

NOTE: Multifocal primary tumours (e.g. bilateral adrenal primary tumours) should be staged according to the greatest extent of disease, as defined above, and followed by a subscript letter M.

[a]The midline is defined as the vertebral column. Tumours originating on one side and crossing the midline must infiltrate to or beyond the opposite side of the vertebral column.

[b]Marrow involvement in stage 4S should be minimal, i.e. less than 10% of total nucleated cells identified as malignant on bone marrow biopsy or on marrow aspirate. More extensive marrow involvement would be considered to be stage 4. The metaiodobenzylguanidine (MIBG) scan (if performed) should be negative in the marrow. Adapted from Brodeur et al. [5].

NB patient cohorts have identified a relative paucity of recurrent alterations [20,24–26].

Initial investigations for NB involve laboratory testing for full blood count, serum electrolytes, liver function and urine catecholamine metabolites [27]. More general biomarkers such as ferritin, lactate dehydrogenase and neuron-specific enolase (NSE) may also be investigated [28]. For suspected NB in the abdomen, ultrasound is the preferred imaging method [29]. A provisional diagnosis is followed up with cross-sectional imaging such as computed tomography or magnetic resonance imaging and confirmed by histological analysis of tumour tissue obtained from a primary tissue biopsy or bone marrow aspirate [29,30].

The treatment algorithm for NB is dependent on risk stratification, which is defined using parameters such as age, disease stage, tumour histopathology, *MYCN* status and DNA ploidy [31]. Low-risk patients often require surgery alone or close observation, since spontaneous regression is frequently observed in this risk group [31]. By contrast, intermediate-risk patients require both surgery and chemotherapy of moderate intensity, and high-risk patients are treated with high-intensity chemotherapy, radiotherapy, surgery and autologous haematopoietic stem cell transplant [31,32]. In addition, high-risk patients receive immunotherapy with anti-GD2 antibodies and cytokines, and differentiation therapy with 13-cis-retinoic acid to eliminate minimal residual disease (MRD) [33].

# 2. Current biomarkers in neuroblastoma

NB is one of few paediatric cancers in which biomarkers are routinely used for diagnosis, prognostication and therapeutic monitoring (table 2).

## 2.1. Urine catecholamines

The majority of neural crest tumours including NB secrete catecholamines [64]. Elevated urinary levels of the catecholamine metabolites vanillylmandelic acid (VMA) and homovanillic acid (HVA) are observed in 90–95% of NB patients at diagnosis [34,35] and a low VMA-to-HVA ratio is associated with poorly differentiated tumours and poor prognosis [36,37]. These metabolites have been used since the 1970s as non-invasive biomarkers to assist in the diagnosis and therapeutic monitoring of patients with NB [38]. A recent study found the combined diagnostic sensitivity of VMA and HVA in NB to be 84% overall [39], though sensitivity is much lower (33–59%) in stage I tumours [36,39]. To facilitate early detection of NB, a screening programme based on urine catecholamine levels in infants aged six months was trialled and later implemented in Japan [65]. However, the programme was terminated upon publication of evidence from screening trials conducted in other countries, which suggested that NB-specific mortality was not reduced among screened subjects [66–68]. Retrospective analyses have determined that screening for NB results in overdiagnosis; screen-detected patients had a tendency to spontaneously regress [69,70] and many of these tumours showed favourable prognostic features at diagnosis [71].

## 2.2. Serum proteins

Serum lactate dehydrogenase (LDH) is used as a tumour biomarker in several malignancies [72], although levels can be elevated in non-malignant conditions such as heart failure, kidney disease, hypothyroidism and anaemia [73]. In NB, elevated serum LDH levels have been shown to confer poor prognosis independent of disease stage in patients with *MYCN* amplification (MNA) [37,40]. Moreover, a recent study identified elevated serum LDH levels as an

**Table 2.** Current biomarkers in NB.

| biomarker | specimen | inference | studies |
|---|---|---|---|
| catecholamine metabolites (HVA, VMA) | urine | diagnostic; prognostic | [34 – 39] |
| lactate dehydrogenase | serum | diagnostic[a]; prognostic | [7,37,40,41] |
| neuron-specific enolase | serum | prognostic | [42 – 46] |
| ferritin | serum | prognostic | [47 – 50] |
| *MYCN* amplification | tissue | prognostic | [11,51 – 53] |
| 1p deletion | tissue | prognostic | [14,54] |
| 11q deletion | tissue | prognostic | [14,55 – 57] |
| 17q gain | tissue | prognostic | [15,54,58,59] |
| *ALK* mutation | tissue | prognostic; therapeutic | [22,23,60 – 62] |
| *ALK* amplification | tissue | prognostic; therapeutic | [60,63] |

[a]Not independent.

independent predictor of poorer event-free (EFS) and overall (OS) survival in patients with metastatic disease older than 18 months at diagnosis [74]. LDH levels are typically normal in the early stages of disease, and significantly rise with tumour burden as the rate of cell turnover increases [7,41]. Thus, serum LDH levels may be used as a surrogate for the real-time monitoring of tumour burden.

NSE is a glycolytic enzyme produced in neural tissues [75]. Serum NSE levels are elevated in neuroendocrine tumours such as carcinoids, islet cell tumours, small-cell lung carcinomas and NB [42,43]. While elevated NSE levels are therefore not specific for NB, they can be used in patients with a confirmed diagnosis to provide prognostic information [42,44,45]. Despite extensive tumour burden, serum NSE levels are significantly lower in stage 4S relative to stage 4 disease [44,46].

NB cell lines and tumours produce and secrete glycosylated ferritin, which can be distinguished from the non-glycosylated form secreted by healthy cells [47]. Elevated levels of serum ferritin are observed in patients with stage 3 or 4 NB, serving as a poor prognostic biomarker independent of age at diagnosis and disease stage [48,49]. Moreover, while ferritin levels in tumour tissue of patients with stage 4 and 4S disease are comparable, serum ferritin levels are only significantly elevated in stage 4 patients, thus enabling discrimination between these two clinical stages with markedly different prognoses [50].

## 2.3. Tumour genetics

Several genetic alterations in NB are routinely detected in diagnostic tissue and serve as important prognostic and therapeutic biomarkers. In addition to *MYCN* as described above, a plethora of segmental chromosome alterations (SCAs) are associated with poor prognosis in NB, notably deletion of 1p and 11q, and gain of 17q; most high-risk tumours carry at least one such alteration. Both 1p36 deletion and 17q gain are associated with MNA and independently correlate with poor outcome [14]. Gain of 17q is the most frequent genetic alteration in NB and is associated with 1p deletion [54]. Loss of 11q is almost mutually exclusive of MNA but is also a feature of high-risk disease and portends an unfavourable prognosis [14,55]. Interestingly, tumours

with 11q loss tend to display numerous SCAs, suggestive of a chromosomal instability phenotype [56,57].

Around 8–10% of sporadic NB tumours present with point mutations in the kinase domain of the full-length ALK receptor [22,23]; this is in contrast to the oncogenic NPM-ALK and EML4-ALK fusions that drive ALK-positive anaplastic large cell lymphoma (ALCL) and non-small cell lung cancer (NSCLC), respectively. *ALK* mutations in NB are found in equal frequencies across all tumour risk groups but appear to be associated with MNA [60], likely representing a cooperative effect between these oncogenes [61]. Additionally, 1–2% of NB tumours harbour genomic amplification of *ALK*, which almost exclusively occurs with MNA given their proximal association at chromosome 2p23-24 [60]. Therefore, *ALK* amplification also tends to afford poor prognosis [63]. ALK alterations serve as important biomarkers in NB because they confer sensitivity to small-molecule kinase inhibitors that are currently undergoing clinical assessment in phase I and II trials [76].

## 3. Emerging tumour-specific circulating biomarkers in neuroblastoma

Tissue-based nucleic acid biomarkers are instrumental to risk classification, prognostication and therapeutic assignment in NB and indeed many other cancers [77]. However, surgical acquisition of tumour tissue is invasive, not amenable to sequential application and potentially subject to sampling bias owing to intratumoral genetic heterogeneity, as highlighted in several recent NB studies [78–80]. Over the past two decades, there has been growing interest in the development of other blood-based tissue biomarker analytes such as cell-free DNA and RNA, circulating tumour cells (CTCs) and, more recently, extracellular vesicles and tumour-educated platelets (figure 1). Collectively, these circulating analytes have been termed a 'liquid biopsy' of cancer and have been reviewed extensively elsewhere [81–83]. The non-invasive nature of blood sampling overcomes several limitations of conventional tissue-based tumour analysis. Whereas invasive biopsy procedures are not amenable to repeated sampling, a liquid biopsy can be taken at multiple time points such as at diagnosis, during treatment and at relapse, capturing the

royalsocietypublishing.org/journal/rsob    Open Biol. **9**: 190056

royalsocietypublishing.org/journal/rsob    Open Biol. 9: 190056

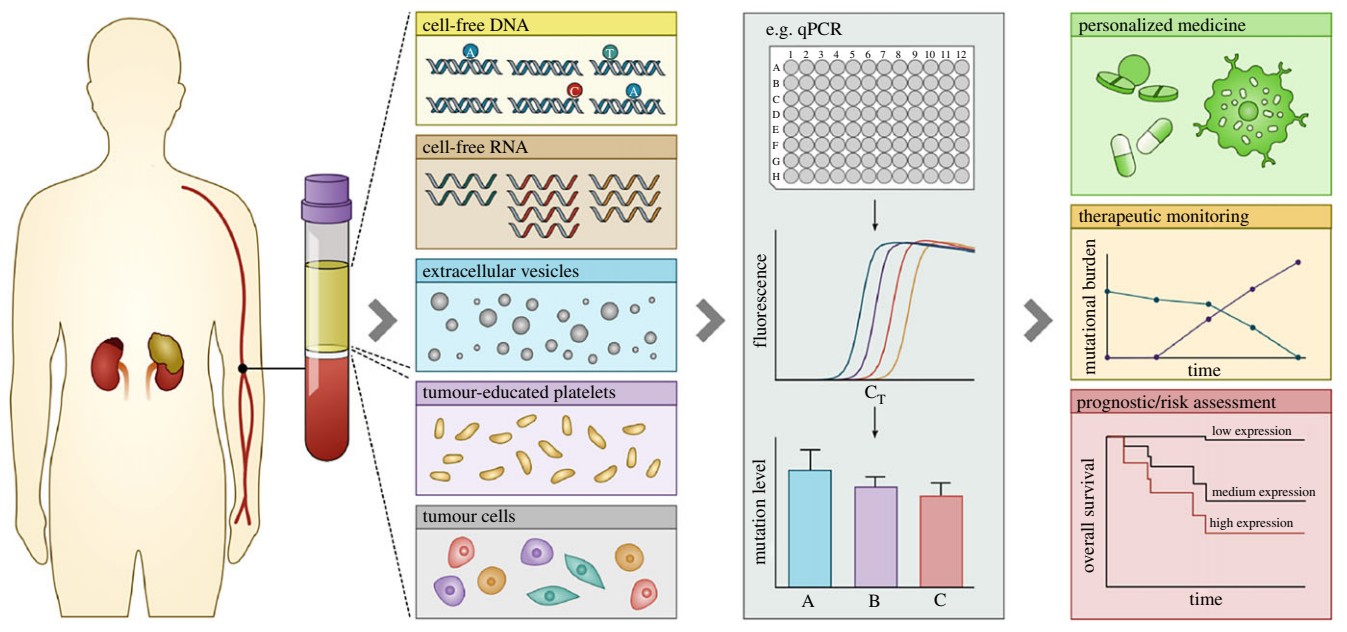

**Figure 1.** The liquid biopsy workflow. In oncology, a liquid biopsy is a minimally invasive alternative to surgical tumour sampling, involving the collection and analysis of a peripheral blood sample. Blood plasma/serum is a source of several circulating analytes with potential clinical utility, namely cell-free DNA, cell-free RNA (mRNA and microRNA), extracellular vesicles (exosomes and microvesicles), tumour-educated platelets (TEPs) and tumour cells. Cell-free nucleic acids are a potential source of tumour-specific genomic alterations such as gene mutations, translocations, copy number alterations (CNAs), DNA methylation and gene expression changes. TEPs are a source of nucleic acids, and extracellular vesicles and tumour cells are a source of nucleic acids and protein. Molecular analysis of circulating biomarkers can impact the clinical management of cancer patients by enabling personalized medicine, monitoring of therapeutic response and assessment of disease prognosis/risk.

genetic and proteomic changes that underlie tumour evolution in real time [84]. Some notable clinical applications of circulating biomarkers include the quantitative real-time polymerase chain reaction (qPCR)-based test recently approved by the US Food and Drug Administration (FDA) for *EGFR* mutations in circulating cell-free DNA from patients with NSCLC, enabling identification of patients likely to respond to EGFR inhibitors [85], and the FDA-approved CELLSEARCH® test for enumeration of CTCs of epithelial origin to aid in the monitoring of patients with metastatic breast, colorectal or prostate cancers [86].

## 3.1. Circulating-free DNA and circulating tumour DNA

DNA was first reported in the circulation 70 years ago [87], but generated little interest until 1977 when levels of circulating free DNA (cfDNA) were shown to be elevated in patients with cancer [88]. Later studies that identified *RAS* mutations in the blood of patients with pancreatic cancer and acute myelogenous leukaemia provided evidence of a significant tumour-derived component of cfDNA, now termed circulating tumour DNA (ctDNA), and these studies led to further exploration of the clinical potential of cfDNA in patients with cancer [89,90]. Basal levels of cfDNA are thought to arise from turnover of normal haematopoietic cells, whereas elevated levels are sometimes observed in cancer and derive from a combination of apoptosis, necrosis and active secretion from tumour cells [91–93]. Indeed, ctDNA is detectable in most patients with NB [94,95]. Moreover, in many solid cancers, including NB [96,97], cfDNA levels have been shown to reflect tumour burden, rising with disease progression and falling after therapy and surgical resection [80,83,95,96]. Thus, cfDNA levels may be a dynamic biomarker for disease

monitoring, though levels can also be elevated in non-malignant conditions [98–100]. In a recent study, cfDNA was shown to be as reliable as NSE and LDH in discriminating NB patients with newly diagnosed ($n = 79$) versus stable ($n = 79$) disease (area under the curve (AUC), 0.953, 0.929 and 0.906, respectively). Moreover, in newly diagnosed patients, elevated plasma cfDNA levels were associated with high-risk disease, advanced tumour stage, MNA, abdominal primary site and three or more metastatic sites [96]. This study did not report a significant association between the DNA integrity index (ratio of long to short cfDNA fragments) and these clinical variables. Patients with NB appear to have high levels of cfDNA at diagnosis, of which a high proportion is tumour derived, i.e. ctDNA [94,101], and reports have demonstrated the detection of numerous genomic alterations, including MNA [80,95,97,101–110], *ALK* mutation [111], 11q deletion [80,95,110,112], 17q gain [80,110,113], and gene methylation [114] in cfDNA of patients with NB at various stages of the clinical course.

### 3.1.1. *MYCN* amplification

Owing to its strong association with high-risk NB, *MYCN* was the first oncogene found to be altered with clinical significance and was consequently the first nucleic acid biomarker in oncology [11,12,51]. Targeted overexpression of N-MYC in neuroectodermal cells is sufficient to induce NB in mice, thus confirming a pathogenic role of *MYCN* in NB [115]. Indeed, MNA remains the strongest indicator of risk in NB, and its levels correlate with aggressive, metastatic behaviour [116,117].

Interphase fluorescence *in situ* hybridization (I-FISH) on biopsy and resected tumour tissue remains the gold standard for assessment of MNA in NB [118]. A number of studies

have demonstrated the potential to detect MNA in serum and plasma from patients with NB at diagnosis across all tumour stages using PCR methodology (table 3), thus raising the prospect of using *MYCN* as a circulating biomarker [102–109]. Using conventional PCR, Combaret *et al.* [102] were the first to show MNA in the sera of 31/32 patients with MNA tumours, with one false positive among 70 patients without MNA and no false positives among 72 healthy patients. Moreover, *MYCN* levels in eight patients were shown to fall after chemotherapy and rise at relapse, and in one patient elevated levels were observed two months before a clinical diagnosis of relapse [102]. A recent study also reported a decrease in plasma *MYCN* copy number one week after surgery in five patients with MNA, proportional to the extent of tumour resection [106]. These observations suggest that *MYCN* in cfDNA may serve as a dynamic biomarker for disease monitoring and early detection of relapse. Indeed, additional studies by Combaret *et al.* and other groups confirmed the high performance of qPCR and digital PCR (dPCR)-based strategies for detecting MNA in serum [103–105,107,108] and plasma [106,109]. While cohort sizes and INSS disease stages were variable between studies, the reported sensitivities and specificities of MNA detection in cfDNA were 84–100% and 95–100%, respectively (table 3). There was also wide inter-study variation in median blood *MYCN* copy numbers from patients with tissue-confirmed MNA (range, 2.56–199.32), likely reflecting the aforementioned differences in cohort characteristics. It is also noteworthy that serum generally yields higher 'apparent' cfDNA levels than plasma [119] due to release of genomic DNA from haematopoietic cells during the clotting process [120–122]. Gotoh *et al.* [103] demonstrated this dilution effect in the context of *MYCN* copy number measurement by spiking white blood cells into serum samples from patients with MNA, resulting in reduced *MYCN*/reference gene ratios. Therefore, delayed isolation of serum could result in significant genomic DNA contamination, thus reducing apparent *MYCN* copy number measurements. For this reason, plasma is more suitable than serum for analysis of gene copy number in cfDNA [119].

A 2005 case report by Combaret *et al.* [104] highlighted the utility of *MYCN* evaluation in cfDNA where tumour biopsy is not possible. A 30-day-old infant presented with stage 4S NB, but thrombocytopenia and hypofibrinogenaemia confounded the collection of tumour tissue. Evaluation of serum *MYCN* by qPCR revealed high-level MNA, and the child was assigned high-intensity chemotherapy. Three weeks after a significant response, blood coagulation parameters had returned to normal and biopsy tissue was obtained, which showed greater than 50 *MYCN* copies by qPCR. The child achieved remission and was disease free 1 year later [104]. It is estimated that around 29% of patients with NB have unknown *MYCN* status [123,124], representing a significant proportion of patients who could potentially benefit from *MYCN* testing in cfDNA.

More recent studies have assessed *MYCN* status in plasma using whole-genome sequencing, whole-genome copy number profiling (i.e. arrayCGH and OncoScan) [80,95,110] and dPCR [97,109]. dPCR is less prone to technical bias than qPCR for copy number measurement because it relies neither on PCR efficiency nor on the availability of a stable reference gene [125]. Aside from its non-invasive and repeatable nature, *MYCN* evaluation in cfDNA offers several other advantages over tissue-based methods. First, it enables rapid determination of *MYCN* status, which is particularly important for patients younger than 18 months at diagnosis where risk grouping is dependent on such information [126,127], and could be implemented in parallel with urine catecholamine tests. Second, the technique is low cost and amenable to a high-throughput format, requiring only DNA extraction from small blood volumes and subsequent PCR amplification of *MYCN*, alone or along with a reference gene such as *NAGK* [103,105–109].

### 3.1.2. Segmental chromosome alterations

Numerous SCAs detected in tumour tissue have been shown to correlate with prognosis in NB by univariate [14,54,128–134] and multivariate [135–137] analyses, and indeed the presence of any SCA is strongly associated with a higher risk of relapse and poorer outcome than in patients with only numerical chromosome alterations [57,138–141]. Several studies have demonstrated the ability to detect recurrent SCAs such as gain of 17q and losses of 11q and 1p in cfDNA isolated from plasma and serum of patients with NB at diagnosis, and have frequently shown high concordances of detection with paired tissue samples [80,110,112,113].

Combaret *et al.* [113] used a duplex qPCR approach to simultaneously amplify genes at 17q.23.1 and 17q25 and calculate copy numbers relative to *TP53* (17p13.1) in cfDNA from patients with NB. With an arbitrary copy number threshold of 1.35, none of 16 cfDNA samples from healthy subjects showed 17q gain, whereas 38% (43/112) of cfDNA samples from patients with NB were positive. Of 58 patients with tissue-confirmed 17q gain, 31 were positive in cfDNA, while 12 of 84 patients with 17q normal tumours showed 17q gain. Consistent with the notion of rising cfDNA levels with disease progression [96,97], diagnostic sensitivity was greater in INSS stage 4 patients (60%) relative to stage 1 and 2 patients (33%), though specificity showed the inverse trend (71.4% versus 88%, respectively) and was, in fact, greater in patients younger than 18 months of age at diagnosis (94.4% versus 71.4%). Indeed, a similar trend was observed upon stratification by *MYCN* status. It is worth noting that the sensitivity of this assay is lower than that reported for detection of MNA in cfDNA (84–100% and 95–100%, respectively), likely due to the masking effect of healthy cfDNA on moderate 17q gains, particularly in patients with low tumour burden [113]. A recent whole-exome sequencing (WES) analysis in NB demonstrated a high concordance between tumour tissue and plasma cfDNA with respect to 17q status; of 19 patients, 12 patients showed 17q gain in both tissue and cfDNA at diagnosis, and one patient was positive for 17q gain in cfDNA only [110]. Similarly, an earlier study by the same group showed a high concordance of 17q gain between primary tumour tissue and cfDNA by analysis with array CGH and OncoScan microarray, respectively. Of 70 patients at diagnosis, 30 showed 17q gain in both tissue and cfDNA, and two patients showed 17q gain in either tissue or cfDNA [80]. Analysis of 17q gain in cfDNA may be useful as a tool for prognostication and therapeutic decision-making alongside cfDNA-based *MYCN* testing, particularly in very young patients in whom tissue-based genomic analyses may not be possible or sufficiently reliable.

Reflecting the clinical importance of 11q loss as a negative prognostic indicator in NB patients without MNA [142],

royalsocietypublishing.org/journal/rsob   Open Biol. **9**: 190056

**Table 3.** Summary of NB studies evaluating MYCN amplification status in peripheral blood using PCR methodology.

| serum/ plasma | PCR method | ref. gene | MNA status (−/+) by INSS disease stage | | | | | | | | | | median blood MYCN ratio by tissue status | | overall sens. (%) | overall spec. (%) | study |
|---|---|---|---|---|---|---|---|---|---|---|---|---|---|---|---|---|---|
| | | | tissue | | | | | serum/plasma | | | | | | | | | |
| | | | 1 | 2 | 3 | 4 | 4S | 1 | 2 | 3 | 4 | 4S | MNA− | MNA+ | | | |
| serum | qPCR | RPPH1 | 14/0 | 10/1 | 8/5 | 33/25 | 5/1 | 14/0 | 10/1 | 9/4 | 32/26 | 5/1 | NR | NR | 97 | 99 | [102] |
| serum | qPCR | NAGK | 22/1 | 18/1 | 7/2 | 18/13 | 5/0 | NR | NR | NR | NR | NR | 0.87 | 199.32 | 100 | 100 | [103] |
| serum | qPCR | IL1B | NR | NR | NR | NR | NR | NR | NR | NR | NR | NR | NR | NR | 89 | 98 | [104] |
| serum | qPCR | NAGK | 24/10[a] | 24/10[a] | 27/16 | 83/41 | 60/6 | 33/1[a] | 33/1[a] | 31/12 | 89/35 | 61/5 | NR | NR | 84 | 100 | [105] |
| plasma | qPCR | NAGK | 16/0 | 4/0 | 7/2 | 6/14 | 1/0 | 16/0 | 4/0 | 7/2 | 6/14 | 1/0 | 0.98 | 26.75 | 100 | 100 | [106] |
| serum | qPCR | NAGK | 38/6[a] | 38/6[a] | 14/12 | 33/38 | 6/1 | 38/6[a] | 38/6[a] | 13/13 | 37/34 | 6/1 | 2.45 | 118.27 | 86 | 95 | [107] |
| serum | qPCR | NAGK | 10/0 | 21/0 | 13/1 | 49/9 | 2/0 | NR | NR | NR | NR | NR | 0.97 | 2.56 | 91 | 98 | [108] |
| plasma | ddPCR | NAGK | NR | NR | NR | NR | NR | NR | NR | NR | NR | NR | 2.7 | 49.2 | 100 | 100 | [109] |

[a]Stages 1 and 2 grouped. ddPCR, droplet digital PCR; INSS, International Neuroblastoma Staging System; MNA, MYCN amplification; NR, not reported; qPCR, quantitative real-time PCR; sens., sensitivity; spec., specificity.

several groups have reported the detection of 11q loss in plasma and serum from patients with NB at diagnosis and relapse [80,110,112]. Yagyu *et al*. [112] used a microsatellite analysis approach to determine allelic status at 11q23 in the serum of 24 patients at diagnosis. Serum allelic intensity scores between tissue-confirmed 11q loss-positive and loss-negative patients did not overlap, and there was full concordance of 11q status between tissue and serum in these patients. Both studies by Chicard *et al*. [80,110] also demonstrated 11q loss in plasma from patients at diagnosis using OncoScan microarray or WES with high concordance between tissue and plasma. The earlier study used OncoScan microarray and array CGH to analyse plasma and tumour DNA of 70 patients, respectively, and achieved a sensitivity of 91% and specificity of 94% [80]. In the later study of 19 patients, paired WES of tissue and plasma demonstrated 100% sensitivity and specificity [110]. Moreover, these studies were able to detect 1p loss in plasma to high concordance with matched tissue, along with other recurrent SCAs such as gains of 1q, 2p and losses of 3p, 4p and 14q [80,110]. Given that SCAs confer poor prognosis in NB and therefore define patients that require more intensive treatment, larger scale studies employing rapid and targeted methodologies for detecting specific SCAs in cfDNA at diagnosis are warranted.

### 3.1.3. *ALK* mutations and amplification

Gain-of-function alterations in *ALK*, namely point mutations and gene amplification, are observed in around 10–12% of patients with NB at diagnosis and tend to afford a poor prognosis [23,60,62]. Recently, small-molecule inhibitors of ALK currently used to treat ALK fusion-positive NSCLC have entered clinical trials for ALK-positive NB and other ALK-positive paediatric cancers [76]. Therefore, genomic alterations of *ALK* may serve as important biomarkers in cfDNA for therapeutic stratification and monitoring of treatment response in the near future. Combaret *et al*. [111] developed dPCR assays for the sensitive detection of point mutations at the two most common mutational hotspots of *ALK* in NB: F1174L (c.3520T > C and c.3522C > A) and R1275Q (c.3824G > A). Among 111 plasma/serum samples obtained from patients with NB at diagnosis, 20 patients were found to be positive for a single mutation and four patients were positive for both c.3522C > A and c.3824G > A. Mutant-to-wildtype *ALK* ratios ranged from 0.15% to 43.7%, likely reflecting different tumour burdens and disease stages within the cohort. Detection of the 3520T > C, 3522C > A and 3824G > A point mutations in cfDNA was achieved with a sensitivity of 100%, 85% and 92%, respectively, and a specificity of 100%, 91% and 98%, respectively. However, the specificity could have been underestimated due to potential spatial sampling bias when obtaining biopsies from genetically heterogeneous tumours [111,143]. *ALK* mutations have also been detected in cfDNA from patients with NB using targeted [97] and whole-exome [110] sequencing approaches, though these studies have used small patient cohorts. Lodrini *et al*. [109] recently developed a dPCR assay for quantification of *ALK* copy number and demonstrated its performance in cfDNA isolated from patients and patient-derived mouse xenografts. Of 10 patients in the study, one patient showed *ALK* gain (copy number greater than or equal to 3) in both cfDNA and tissue by dPCR and two patients showed *ALK* gain in cfDNA but not in tissue. The latter observation could have been due to

*ALK* gains in subclones of the primary tumours or in metastatic sites undetected at diagnosis [109,143].

### 3.1.4. DNA methylation

Silencing of tumour suppressors by hypermethylation of promoter CpG islands is a key epigenetic event in tumorigenesis [144] and a promising circulating biomarker in diverse cancer types [145]. Several tumour suppressors are frequently silenced by hypermethylation in NB [146]. CpG island hypermethylation of the RAS effector protein *RASSF1A* is found in the majority of NB tumours at all clinical stages [146–149] and is not associated with prognostic factors such as *MYCN* status or age at diagnosis [148,150]. By contrast, *RASSF1A* hypermethylation in the pre-treatment serum of patients with NB has been shown to be significantly associated with age > 12 months at diagnosis and advanced (INSS stage 4) MNA disease [148]. In this study, Misawa *et al*. [148] used methylation-specific PCR and detected serum *RASSF1A* hypermethylation in 25% (17/68) of patients, demonstrating its utility as a poor prognostic factor comparable to that of MNA by univariate analysis. The same group also investigated the prognostic potential of serum *DCR2* hypermethylation, given its association with poor outcome in primary NB tumours [114,151]. *DCR2* hypermethylation was associated with tumour stage, independent of *MYCN* status, and patients with this alteration had a poorer 5 year EFS, which was particularly significant among patients without MNA [114]. Moreover, hypermethylation levels were found to decrease towards disease remission and become elevated at relapse, thus demonstrating the potential of serum *DCR2* hypermethylation as a dynamic biomarker for prognostication and therapeutic monitoring in NB [114].

## 3.2. Circulating microRNA

MicroRNAs (miRNAs) are a family of endogenous, short (20–25 nt) non-coding RNA molecules that provide post-transcriptional regulation of gene expression [152]. Dysregulation of miRNAs is widely observed in cancer, often caused by mechanisms such as deletion, amplification and changes in gene expression [152]. Over the past decade, miRNAs have been investigated in plasma and serum as non-invasive biomarkers for diagnosis, therapeutic stratification and prognostication across diverse cancer types [153].

Many known oncogenic and tumour-suppressive miRNAs have been shown to be aberrantly expressed in primary NB tumours, with a particular focus on those targeting *MYCN* [154–156] among other genes implicated in NB pathogenesis [157]. In addition, a limited number of studies have demonstrated the roles of specific miRNAs in mediating chemoresistance in NB cell lines and tumours [157]. However, few studies have investigated the expression and clinical utility of circulating miRNAs in NB. Murray *et al*. [158] undertook a global ($n = 741$) reverse transcription (RT)-qPCR-based analysis of miRNA expression in diagnostic serum from 33 paediatric cancer patients and identified a unique expression profile for each tumour type. Sera from NB patients with MNA ($n = 2$) showed overexpression of miR-124-3p, miR-9-3p, miR-218-5p, miR-490-5p and miR-1538, consistent with a previous study demonstrating *MYCN* status as a determinant of global miRNA profiles in NB tissue [159]. Moreover, miR-9 is known to be induced by *MYCN* and its high expression in

NB is associated with MNA and metastatic disease [160]. In a recent study, whole-miRNome profiling was conducted in sera from mice bearing favourable, non-metastatic NB xenografts and mice with high-risk, metastatic disease [161]. The authors identified a circulating miRNA signature of high-risk disease, comprising high and low expression of 34 and 46 miRNAs, respectively. Individual miRNAs were functionally validated by analysing expression of their putative target proteins at distant metastatic sites from the high-risk model, and three miRNAs (miR-381, miR-548h and miR-580) were identified as showing greater than 10-fold increased expression in sera from this model, thereby serving as putative biomarkers of high-risk disease for clinical investigation [161]. Another recent study investigated expression of miRNAs in pooled sera from patients with low- and high-risk NB, identifying 743 well-expressed miRNAs [162]. The authors then evaluated expression of these miRNAs in sera from a cohort of 141 patients with NB at diagnosis, identifying tumour stage as the greatest determinant of variance in miRNA levels. Nine miRNAs that showed significantly different expression between patients with low- and high-risk disease were further investigated, and their expression levels were found to increase with tumour stage. Moreover, in mice xenografted with NB tumours, serum expression levels of all nine miRNAs were found to increase with tumour load, an observation subsequently confirmed by longitudinal blood sampling in five patients with high-risk metastatic NB. Interestingly, expression of the nine miRNAs in primary NB tumours was not significantly different between disease stages, leading the authors to conclude that differential expression between disease stages in sera most likely reflects tumour burden and therefore metastatic status [162].

While these studies have provided proof of principle that circulating miRNA profiles can distinguish between favourable and high-risk NB, candidate miRNA biomarkers must be evaluated in large, prospective patient cohorts before their clinical utility can be realized. Moreover, given that miRNA profiles associated with chemoresistance have been identified in NB, further investigation of these miRNAs as circulating biomarkers for therapeutic monitoring is warranted [157].

## 3.3. CTCs and CTC-derived mRNA

CTCs are malignant cells that disseminate into the bloodstream from primary or metastatic tumours and are responsible for seeding metastatic growth [163]. Together with disseminated tumour cells in bone marrow, CTCs are surrogate markers of sub-clinical metastasis (MRD), in which small numbers of tumour cells persist after therapy in patients in remission, often leading to clinical relapse [164]. CTCs are rare, comprising as few as one cell per billion haematological cells, and this has presented a major challenge to their isolation and molecular characterization. Recent advancements in immunological and size-based enrichment methods have enabled enumeration and molecular analysis of CTCs in peripheral blood isolated from patients with diverse cancer types [165]. CTCs are currently under investigation as biomarkers for diagnosis, therapeutic monitoring, prognostication and assessment of relapse risk [166]. The presence of CTCs in patients with NB was first demonstrated through the establishment of NB cell lines during *in vitro* culture of peripheral blood samples from patients with disseminated disease [167,168]. These studies highlighted the potential for tumour cell contamination in stem cell harvests from peripheral blood, a concern supported by a later study demonstrating the clonogenic properties of NB CTCs *in vitro* [169].

### 3.3.1. CTC detection methods

The first prospective molecular analyses of CTCs employed immunocytochemical methods with monoclonal antibodies against neuronal cell markers such as CD56 (NCAM), CD90 (Thy-1) and GD2 [168,170,171]. CTCs were detected in patients with metastatic disease at diagnosis [168,170,171], during therapy [170,171] and at relapse [168]. In one study, the presence of CTCs in patients with metastatic disease during therapy was found to be an indicator of disease relapse [170]. Subsequent studies have used RT-PCR-based methodologies for the indirect detection of CTCs, targeting mRNA with neuron-specific expression such as *UCH-L1* (PGP9.5) [172–175], *TH* (tyrosine hydroxylase) [173–207], *GALGT* (GD2 synthase) [200,204,208], *DCX* (doublecortin) [187,193, 199,203–205,207], *DDC* (DOPA decarboxylase) [192,193,198, 202,204,205] and *PHOX2B* [187,193,202,204,205,207]. Mattano *et al.* [172] evaluated the performance of an RT-PCR assay targeting *UCH-L1*, reporting a 100-fold increase in sensitivity relative to immunocytochemical assays, with the ability to detect a single CTC among $10^7$ peripheral blood mononuclear cells (PBMCs) versus 1 or 2 CTCs among $10^5$ PBMCs by immunocytochemistry. Subsequent RT-qPCR analyses targeting other markers such as *TH* and *GALGT* have demonstrated variable sensitivities, with detection limits ranging from 1 CTC in $10^3$ to 1 CTC in $10^7$ PBMCs [174,176,178,190, 194–202,209].

### 3.3.2. Association of CTCs with clinical features

As with immunocytochemical analyses, RT-PCR-based studies have consistently identified NB-specific mRNA in peripheral blood at diagnosis in patients with metastatic disease and in a fraction of patients with localized and stage 4S disease (table 4). Diagnostic *TH* mRNA levels and CTC counts are typically higher in patients with more advanced metastatic disease [185,204,206,212] and in patients with high-risk disease [206,212], as observed in other solid malignancies [166]. However, NB-specific mRNA levels and CTC counts do not correlate with *MYCN* status [182,184,206,207,212]. In addition to serving as a potential diagnostic biomarker, there is a wealth of evidence supporting the prognostic value of NB-specific mRNA, with numerous studies reporting a correlation between mRNA levels and survival outcomes (table 4). High transcript levels of *TH* and other NB-specific genes at diagnosis have been associated with poor OS and/or EFS in patients with localized and metastatic disease, independent of risk status [187,193,194,197,203,206,208]. Similarly, high CTC counts ($\geq$ 10 cells per 4 ml blood) have recently been shown to correlate with poor OS [212]. Among patients with high-risk disease, high expression of *TH* and *PHOX2B* mRNA at diagnosis has been shown to correlate with remarkably poorer outcome, thus identifying a subset of patients with ultra-high-risk disease who may benefit from novel treatment approaches [187].

Detection of CTCs or NB-specific mRNA in peripheral blood of patients in remission following completion of therapy may indicate MRD and is associated with poorer

**Table 4.** Summary of studies reporting direct and indirect detection of CTCs in peripheral blood of patients with NB. CTC, circulating tumour cell; EFS, event-free survival; FISH, fluorescence *in situ* hybridization; ICC, immunocytochemistry; MRD, minimal residual disease; NB, neuroblastoma; ND, not determined; RT-(q)PCR, reverse transcription (quantitative real-time) PCR; INSS, International Neuroblastoma Staging System; OS, overall survival; PFS, progression-free survival.

| detection method(s) | marker(s) | sensitivity (tumour cell *n*/non-tumour cell *n*) | main observation(s) in peripheral blood | study |
|---|---|---|---|---|
| ICC | NCAM, unknown NB-specific antigen | ND | CTCs detected in patients with metastatic disease at diagnosis and relapse. | Lanino *et al.* [168] |
| ICC | GD2, Thy-1, unknown NB-specific antigen | $2/10^5$ | CTCs detected in patients with metastatic disease at diagnosis and during therapy. The presence of CTCs in patients during therapy associated with disease relapse. | Moss & Sanders [170] |
| ICC | GD2, Thy-1, unknown NB-specific antigen | ND | CTCs detected in patients with metastatic disease at diagnosis. CTCs detected during therapy in one patient. | Sanders *et al.* [171] |
| ICC; RT-PCR | ICC: NCAM, GD2, Thy-1; RT-PCR: *UCH-L1* | ICC: ND; RT-PCR: $1/10^7$ | CTCs/NB-mRNA detected in patients with localized and metastatic disease, and in patients with no evidence of disease. RT-PCR more sensitive than ICC, though analytical sensitivity of ICC not determined. | Mattano *et al.* [172] |
| ICC | NCAM, GD2, Thy-1 | ND | CTCs detected in patients with metastatic disease at diagnosis and during therapy. Around half of ICC-positive blood samples and several negative samples showed clonogenic growth of NB cells *in vitro*. | Moss *et al.* [169] |
| RT-PCR | *TH* | $1 - 10/10^6$ | *TH* mRNA detected in patients with metastatic disease at diagnosis, with some patients undetectable 6–8 weeks into therapy. *TH* mRNA detected in patients with relapsed disease and patients in remission, one of whom later relapsed. | Burchill *et al.* [176] |
| RT-PCR | *TH* | ND | *TH* mRNA detected in patients with metastatic disease at diagnosis, with some patients undetectable 6–8 weeks into therapy. *TH* mRNA detected in patients with relapsed disease and patients in remission, some of whom later relapsed. Some patients in remission and with no detectable *TH* mRNA eventually relapsed. | Burchill *et al.* [177] |
| RT-PCR | *TH* | $1/10^5 - 10^6$ | *TH* mRNA detected in patients with localized and metastatic disease, including INSS stage 4S patients. | Miyajima *et al.* [178] |
| RT-PCR | *TH* | ND | *TH* mRNA detected in patients with localized and metastatic disease, and in patients in remission. Significantly higher *TH* mRNA levels in samples taken before complete remission. | Miyajima *et al.* [179] |
| RT-PCR | *TH* | ND | *TH* mRNA detected in a patient with metastatic disease and undetectable by conventional histology. | Kuroda *et al.* [180] |
| RT-PCR | *TH, UCH-L1* | ND | NB-mRNA detected in patients of unknown tumour stage, some of whom tested negative following treatment. | Gao *et al.* [175] |

(*Continued.*)

**Table 4.** (*Continued.*)

| detection method(s) | marker(s) | sensitivity (tumour cell $n$/non-tumour cell $n$) | main observation(s) in peripheral blood | study |
|---|---|---|---|---|
| RT-PCR | GAGE, TH | GAGE: 1/10$^6$; TH: ND | GAGE mRNA, but not TH mRNA, detected in some patients with metastatic disease at diagnosis. | Cheung & Cheung [190] |
| RT-PCR | TH, UCH-L1 | TH: 10$^{-4}$ μg RNA; UCH-L1: 10$^{-6}$ μg RNA | NB-mRNA detected at diagnosis in patients aged 1 year or older with metastatic disease. RT-PCR detection of UCH-L1 more sensitive than that for TH. | Yanagisawa et al. [173] |
| ICC | GD2 | 1/10$^6$ | CTCs detected in patients with metastatic disease and INSS stage 4S disease at diagnosis. | Faulkner et al. [209] |
| RT-PCR | TH | 1 cell/2 ml blood | TH mRNA detected in patients with metastatic disease at diagnosis. | Burchill et al. [191] |
| RT-PCR | TH | ND | TH mRNA detected in patients with metastatic disease at diagnosis. | Kuroda et al. [181] |
| RT-PCR | TH | 1/10$^6$ | Detection of TH mRNA in patients older than 1 year with metastatic disease associated with poorer OS and EFS. Detection of TH mRNA in patients with no evidence of disease associated with increased risk of relapse and death. | Burchill et al. [194] |
| ICC; RT-PCR | ICC: GD2; RT-PCR: TH, CHGA | GD2: 1/10$^6$; TH: 1/10$^3$; CHGA: 1/10$^6$ | TH RT-PCR less sensitive than CHGA RT-PCR and GD2 ICC. | Pagani et al. [195] |
| RT-qPCR | TH | 70 transcripts/1 ml blood | At diagnosis, TH mRNA levels significantly higher in patients with metastatic/INSS stage 4S disease than limited disease. Some patients positive for TH mRNA before, during and 24 h after surgery. | Träger et al. [185] |
| RT-qPCR | TH | 1/10$^6$ | TH mRNA detected in patients with localized and metastatic diagnosis, and in a patient after seven months of chemotherapy. | Lambooy et al. [196] |
| RT-qPCR | TH | 1/10$^6$ | TH mRNA detected at diagnosis, during chemotherapy and at relapse in patients aged 1 year or older with metastatic disease. High frequency of tumour cell contamination in autologous PBSC harvests; high TH mRNA levels associated with poorer survival. | Tchirkov et al. [197] |
| ICC; RT-PCR | ICC: GD2; RT-PCR: TH, UCH-L1 | GD2: 1/10$^6$; TH: 1/10$^5$; UCH-L1: 1/10$^6$ | Detection of CTCs by GD2 ICC at diagnosis in patients with localized disease correlated with unfavourable outcome. No correlation of TH mRNA status with outcome in patients with metastatic disease. GD2 ICC showed greatest accuracy and UCH-L1 RT-PCR showed poor accuracy. | Corrias et al. [174] |
| RT-PCR | GALGT | 1/10$^6$ | GALGT mRNA detected in patients with metastatic disease at diagnosis and associated with higher rates of relapse and death. | Cheung et al. [208] |
| RT-PCR | GAGE | ND | GAGE mRNA detected in high proportion of healthy subjects; thus, GAGE mRNA not specific for NB. | Oltra et al. [210] |

(*Continued.*)

**Table 4.** (*Continued*.)

| detection method(s) | marker(s) | sensitivity (tumour cell $n$/non-tumour cell $n$) | main observation(s) in peripheral blood | study |
|---|---|---|---|---|
| RT-PCR | *DDC, TH* | $1/10^6$ | *DDC* and *TH* mRNA detected at similar frequencies in patients with metastatic disease at diagnosis. | Bozzi *et al.* [198] |
| RT-PCR | *DCX, TH* | $1/10^5$ | *DCX* and *TH* mRNA detected at similar frequencies in patients with high-risk disease at diagnosis. High correlation between *DCX* and *TH* mRNA levels in positive samples. | Oltra *et al.* [199] |
| RT-qPCR | *TH* | ND | *TH* mRNA detected in patients with localized and metastatic disease at diagnosis. Detection of *TH* mRNA after therapy associated with poorer prognosis. | Parareda *et al.* [183] |
| RT-PCR | *TH* | ND | *TH* mRNA detected in patients with metastatic disease at diagnosis, and all deaths in mRNA-positive patients related to systemic metastases. No association of *TH* mRNA positivity with *MYCN* status. | Kuroda *et al.* [184] |
| RT-PCR | *TH, GALGT, ELAVL4* | *TH*: $1/10^6$; *GALGT*: $1/10^4$; *ELAVL4*: $1/10^6$ | Persistence of high *ELAVL4* mRNA levels during therapy has prognostic value in patients with metastatic disease. *GALGT* RT-PCR less sensitive than that for *TH* or *ELAVL4*. | Swerts *et al.* [200] |
| RT-qPCR | *TH* | $1/10^6$ | For CTC detection, blood should be collected into PAXgene or EDTA tubes and stored for 48 h or less. Implementation of standardized protocols increased sensitivity of *TH* RT-PCR from 58% to 90%. | Viprey *et al.* [201] |
| RT-PCR | *TH* | ND | *TH* mRNA detected in patients with metastatic disease during treatment, but not in patients with stage 3 disease. No association of *TH* mRNA with *MYCN* status. All mRNA-positive patients died of disseminated disease. | Kuroda *et al.* [211] |
| RT-qPCR | *TH, DDC, GD2S* | ND | High NB-mRNA levels associated with poor prognosis in patients with various disease stages. RT-qPCR for *GD2S* less specific than that for *TH* and *DDC*. | Träger *et al.* [192] |
| RT-qPCR | *TH, DDC, DBH, PHOX2B* | *TH*: $1/10^6$; *DDC*: $1/10^6 – 10^7$; *DBH*: $1/10^7$; *PHOX2B*: $1/10^6 –10^7$ | Panel of RT-qPCR markers more sensitive than *PHOX2B* alone in detecting MRD. | Stutterheim *et al.* [202] |
| RT-qPCR | *TH, DCX* | ND | *TH* and/or *DCX* mRNA detected in some patients with localized disease and associated with poorer EFS but not OS. | Yáñez *et al.* [203] |
| RT-qPCR | *TH, GALGT, DDC, DCX, ELAVL4, STX, PHOX2B* | ND | All markers except *PHOX2B* expressed in healthy donors. TH mRNA levels significantly lower in patients with localized versus metastatic disease. Weak correlation between detection of different markers in patients with localized disease, but perfect agreement between *TH* mRNA and *GALGT* mRNA in patients with metastatic disease. | Corrias *et al.* [204] |

(*Continued*.)

royalsocietypublishing.org/journal/rsob *Open Biol.* **9**: 190056

**Table 4.** (Continued.)

| detection method(s) | marker(s) | sensitivity (tumour cell n/non-tumour cell n) | main observation(s) in peripheral blood | study |
|---|---|---|---|---|
| RT-qPCR | CHRNA3, CRMP1, DBH, DCX, DDC, GABRB3, GAP43, ISL1, KIF1A, PHOX2B, TH | ND | Sensitivity of MRD detection is greater with multiple RT-qPCR markers. | Hartomo et al. [205] |
| RT-qPCR | TH, PHOX2B, DCX | ND | High levels of TH, PHOX2B or DCX mRNA at diagnosis in patients with metastatic disease strongly associated with poorer EFS and OS. High levels of TH and PHOX2B at diagnosis identify patients with ultra-high-risk disease who may benefit from novel treatment approaches. | Viprey et al. [187] |
| RT-qPCR | TH | ND | Higher TH mRNA levels at diagnosis in high-risk patients and patients with metastatic disease, but no association with MYCN status. Expression of TH mRNA at diagnosis correlated with poorer 5 year PFS, both independent of risk stage and also when assessed in high-risk patients. No significant correlations with EFS were made after induction therapy. | Lee et al. [206] |
| RT-qPCR | CHGA, DCX, DDC, PHOX2B, TH | ND | NB-mRNA detectable in patients with relapsed/refractory disease. Lower RT-qPCR $\Delta C_T$ values correlated with poorer PFS, independent of disease stage or MYCN status. | Marachelian et al. [193] |
| RT-qPCR | TH, PHOX2B, DCX | ND | Levels of PHOX2B and DCX mRNAs, but not TH mRNA, in the highest tertile were associated with shorter EFS in a cohort of high-risk patients with metastatic disease. Correlation of mRNA and EFS was independent of MYCN status. Prognostic power of PHOX2B mRNA lost when combined with TH mRNA. | Corrias et al. [207] |
| FISH | CD45, CEP8, DAPI | ND | CTCs identified by depletion of non-tumour cells with magnetic beads coated with anti-CD45 antibodies, followed by analysis of non-bound cells with CEP8 probe and DAPI. CTC = CD45-negative, DAPI-positive, CEP8 $\geq$ 3 spots. Significantly more CTCs identified in high-risk patients and patients with metastatic disease. Patients with greater than or equal to 3 CTCs per 4 ml blood had greater likelihood of having metastases, and patients with greater than or equal to 10 per 4 ml blood had poorer OS. No correlation of CTC count with MYCN status. Higher CTC counts in patients with partial response to therapy or progressive disease relative to patients with complete response. | Liu et al. [212] |

prognosis in NB [183]. Consistent with the eradication of systemic disease, several studies have shown that mRNA may decrease and/or become undetectable during and after chemotherapy [175–177,212]. CTCs or detectable levels of mRNA that remain upon completion of chemotherapy or after surgical resection are associated with an increased risk of relapse [170,176,177,183,185,197,200]. Thus, mRNA in peripheral blood has potential as a dynamic biomarker for monitoring response to therapy and evaluation of relapse risk in patients with NB.

Several groups have demonstrated an increase in sensitivity of MRD detection in peripheral blood and bone marrow by evaluating multiple mRNAs simultaneously, which is perhaps expected given the genetic heterogeneity of NB [193,202,204,205]. However, some mRNAs such as *GALGT* have been shown to lack sensitivity [200] and contribute false positivity [189,192], and hence should be validated individually before incorporation into multi-marker panels. Corrias *et al.* [204] evaluated a panel of seven mRNAs and found no correlation of any individual mRNA with specific clinical features in patients with localized disease. Moreover, a high rate of false positivity was observed upon combined analysis of all mRNAs, leading the authors to conclude that multi-marker analysis may not offer benefit over analysis of a single mRNA in patients with low tumour burden [204].

Towards clinical implementation of mRNA-based MRD analysis in patients with NB, Viprey *et al.* [201] coordinated an initiative to standardize the methodology for detection of NB cells by RT-qPCR. Original methodologies on blood collection, RNA isolation and PCR protocols were evaluated from several reference laboratories across Europe and a standardized, quality-controlled protocol was devised. This protocol led to an increase in sensitivity of *TH* mRNA detection by RT-qPCR from 58% to 90% [201] and has since been implemented by the same group for MRD detection in patients with metastatic disease enrolled on the international phase 3 HR-NBL-1/SIOPEN trial [187,207]. In the first of two studies, Viprey *et al.* [187] reported that levels of *TH*, *PHOX2B* and *DCX* mRNAs in peripheral blood (and bone marrow) of patients at diagnosis and at the end of induction therapy were predictive of EFS. A recently published follow-up study evaluated levels of *TH*, *PHOX2B* and *DCX* mRNAs in patients < 18 months of age at diagnosis, since this age group was not sufficiently represented in the first study. In these patients, levels of *PHOX2B* and *DCX* mRNA, but not *TH* mRNA, in the highest tertile at diagnosis were associated with shorter EFS, and *PHOX2B* mRNA alone showed prognostic power in patients during the first year of follow-up [207]. Studies into MRD detection in patients with NB (table 4) have clearly demonstrated the potential for NB-mRNAs in peripheral blood as non-invasive predictive and prognostic biomarkers within defined patient subsets. The clinical significance of NB-mRNAs in peripheral blood is worthy of further exploration in comparison with established biomarkers in large multi-centre trials using standardized protocols before integration into the clinic [201,213].

## 4. Clinical implementation of circulating biomarkers in neuroblastoma

A major obstacle to the discovery and validation of clinically applicable biomarkers in NB, and indeed paediatric cancers

in general, is the scarcity of patient material available for correlative research. There is a tendency to take core needle biopsies rather than tissue sections, which limits the number of tests that can be performed. In addition, protocols used for blood collection, plasma/serum isolation and specimen storage may not permit optimal recovery of biomarker analytes such as nucleic acids [119]; these methodological factors may be a significant cause of inter-study variability, as exemplified by the choice of plasma or serum for analysis of MNA in blood [102–109]. Furthermore, retrospective biomarker analysis requires access to patient clinicopathological data, which is not always possible or easy unless in the context of a clinical trial. Indeed, inferences made from prospective biomarker analyses in patients enrolled on clinical trials are often limited due to small cohort sizes that result from stratification of patients into treatment arms or into groups based on clinicopathological features such as age, risk status and disease stage [201,207]. Despite these limitations, it has been possible to develop prognostic assays.

There is sufficient retrospective evidence to indicate that analysis of MNA in blood by qPCR can determine *MYCN* status in the majority of NB patients with advanced disease, thus serving as a biomarker for prognostication and potentially response to therapy [102–109]. The most obvious benefit of blood-based *MYCN* analysis is the rapid determination of risk status in patients at diagnosis, enabling immediate application of appropriate treatment. Blood-based *MYCN* assessment should now be incorporated into large-scale prospective trials in patients at diagnosis, during/after induction therapy and at relapse. A limited number of retrospective studies have demonstrated successful detection of various SCAs in plasma and serum of patients with advanced disease [80,110,112,113]. Given that SCAs are currently used as tissue-based indicators of poor prognosis, further analysis of their detectability in blood in larger patient cohorts is warranted, perhaps in tandem with *MYCN* analysis [113]. The relatively 'quiet' nature of NB genomes has not provided a plethora of potential aberrations for the development of further blood-based assays, which has further confounded the implementation of this approach to patient management.

To date, ALK is the only druggable gene product that is frequently mutated in NB, and small-molecule inhibitors of ALK currently approved for the treatment of ALK-positive NSCLC are undergoing clinical assessment in patients with NB and other ALK-positive paediatric malignancies. However, a wealth of preclinical studies and a few clinical case reports in patients with NB are highlighting the issue of therapeutic resistance, thus compromising the long-term efficacy of these compounds [76]. The ability to detect *ALK* mutations in the blood at diagnosis and relapse, and to monitor mutations during treatment, would enable the rapid assignment of ALK inhibitors to eligible patients and to monitor treatment efficacy in real time. In particular, given that many children are too unwell for re-biopsy on relapse, an assay that can determine the identity of the *ALK* mutation on relapse would be highly beneficial in the clinic. For example, emergence of secondary *ALK* mutations in the blood would indicate resistance and may provide the rationale for switching to other structurally related compounds with distinct resistance profiles. Combaret *et al.* [111] have demonstrated the detection of mutations at the F1174 and R1275 hotspots of ALK in cfDNA of patients at diagnosis

royalsocietypublishing.org/journal/rsob    Open Biol. 9: 190056

royalsocietypublishing.org/journal/rsob Open Biol. 9: 190056

using a droplet digital PCR (ddPCR) methodology [111], and several other groups have detected point mutations in *ALK* by next-generation sequencing in small patient cohorts [97,110]. The utility of *ALK* mutations in cfDNA as therapeutic biomarkers will only be realized upon integration of cfDNA analysis into large, well-designed prospective trials that include an ALK inhibitor treatment arm, several of which are currently underway [214,215].

Another clinical application of circulating biomarkers is the detection of MRD, a strong predictor of relapse in cancer [164]. To this end, a variety of MRD detection methods have been developed based on direct and indirect detection of tumour cells in peripheral blood [216]. Early studies in patients with NB employed immunocytochemical methods to detect CTCs, and later studies revealed the increased sensitivity offered by qualitative and quantitative RT-PCR-based strategies targeting NB-specific mRNAs (table 4). Numerous retrospective analyses have amassed a substantial evidence base around the utility of qualitative and quantitative RT-PCR for detection of rare CTCs in NB, with a sensitivity of a single tumour cell in up to 10 million normal haematopoietic cells [172,202]. Of particular clinical interest is the ability of NB-specific mRNA levels in blood to predict OS and/or EFS outcomes among patients with metastatic disease, as determined in a series of small-cohort studies [183,192,193,197, 200,203,206] and more recently in larger studies of patients enrolled on the HR-NBL-1/SIOPEN trial [187,207]. RT-qPCR-based MRD analysis must now be incorporated into well-powered prospective clinical trials to assess the predictive and prognostic value of NB-specific mRNA levels in blood, both independently and in combination with established biomarkers.

## 5. Summary

The poor prognosis of patients with high-risk NB necessitates the development of biomarkers to facilitate therapeutic stratification, prognostication and assessment of relapse risk. There is substantial evidence from retrospective studies to warrant investigation of MNA and CTC-derived mRNAs in peripheral blood of patients enrolled on large-scale prospective trials to confirm the clinical utility of these biomarkers in risk stratification and prognostic assessment, respectively. The detection of ALK mutations in cfDNA of patients enrolled on current ALK inhibitor trials will establish the feasibility of real-time non-invasive monitoring of treatment response and detection of resistance.

Data accessibility. This article has no additional data.
Competing interests. We declare we have no competing interests.
Funding. We received no funding for this study.

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
