## [Reviewer comments · Open Biology]

Review History

RSOB-19-0056.R0 (Original submission)

Review form: Reviewer 1

Recommendation

Accept with minor revision (please list in comments)

Are each of the following suitable for general readers?

- a) **Title**
Yes
- b) **Summary**
Yes
- c) **Introduction**
Yes

Is the length of the paper justified?

Yes

Should the paper be seen by a specialist statistical reviewer?

No

Is it clear how to make all supporting data available?

Not Applicable

Is the supplementary material necessary; and if so is it adequate and clear?

Not Applicable

Do you have any ethical concerns with this paper?

No

Comments to the Author

This is a very comprehensive and well-written review of circulating biomarkers in neuroblastoma which would benefit from some minor revision to make it suitable for a readership which might not be so familiar with the neuroblastoma field.

1. The INSS staging system is so central to the discussion of each biomarker, that it deserves more explanation in the introduction. A table would be useful.
2. ALK (and ATRX) should be defined at the first mention in section 1. For the readership of this journal it could be useful to compare and contrast the nature of ALK activation in neuroblastoma as opposed to other malignancies (lung cancer) at the first opportunity.
3. Figure 1 is very informative (although the text in the legend requires formatting), and although the authors state that liquid biopsies have been reviewed extensively elsewhere, for the Open Biology readers it would be useful to give some key examples of clinical translation for the different types of circulating biomarkers in other malignancies.
4. Section 3.3 is very long. Subtitles here would be welcome.
5. "et al" is occasionally italicised, but mostly not.
6. Section 4 is a good translational summary and discussion which could be extended to provide further "constructive discussion and/or critique of the field" (as per Open Biology guidelines). The need for a final further summary (section 5) is questionable.

Decision letter (RSOB-19-0056.R0)

10-Apr-2019

Dear Dr Trigg

We are pleased to inform you that your manuscript RSOB-19-0056 entitled "Opportunities and challenges of circulating biomarkers in neuroblastoma" has been accepted by the Editor for publication in Open Biology. The reviewer has recommended publication, but also suggest some minor revisions to your manuscript. Therefore, we invite you to respond to the reviewer's comments and revise your manuscript.

Please submit the revised version of your manuscript within 7 days. If you do not think you will be able to meet this date please let us know immediately and we can extend this deadline for you.

- 1) A text file of the manuscript (doc, txt, rtf or tex), including the references, tables (including captions) and figure captions. Please remove any tracked changes from the text before submission. PDF files are not an accepted format for the "Main Document".
- 2) A separate electronic file of each figure (tiff, EPS or print-quality PDF preferred). The format should be produced directly from original creation package, or original software format. Please note that PowerPoint files are not accepted.
- 3) Electronic supplementary material: this should be contained in a separate file from the main text and meet our ESM criteria (see <http://royalsocietypublishing.org/instructions-authors#question5>). All supplementary materials accompanying an accepted article will be treated as in their final form. They will be published alongside the paper on the journal website and posted on the online figshare repository. Files on figshare will be made available approximately one week before the accompanying article so that the supplementary material can be attributed a unique DOI.

Online supplementary material will also carry the title and description provided during submission, so please ensure these are accurate and informative. Note that the Royal Society will not edit or typeset supplementary material and it will be hosted as provided. Please ensure that the supplementary material includes the paper details (authors, title, journal name, article DOI). Your article DOI will be 10.1098/rsob.2016[last 4 digits of e.g. 10.1098/rsob.20160049].

- 4) A media summary: a short non-technical summary (up to 100 words) of the key findings/importance of your manuscript. Please try to write in simple English, avoid jargon, explain the importance of the topic, outline the main implications and describe why this topic is newsworthy.

Images

Data-Sharing

It is a condition of publication that data supporting your paper are made available. Data should be made available either in the electronic supplementary material or through an appropriate

repository. Details of how to access data should be included in your paper. Please see <http://royalsocietypublishing.org/site/authors/policy.xhtml#question6> for more details.

Data accessibility section

Sincerely,

The Open Biology Team
<mailto:openbiology@royalsociety.org>

Reviewer's Comments to Author:

Referee:

Comments to the Author(s)

This is a very comprehensive and well-written review of circulating biomarkers in neuroblastoma which would benefit from some minor revision to make it suitable for a readership which might not be so familiar with the neuroblastoma field.

1. The INSS staging system is so central to the discussion of each biomarker, that it deserves more explanation in the introduction. A table would be useful.
2. ALK (and ATRX) should be defined at the first mention in section 1. For the readership of this journal it could be useful to compare and contrast the nature of ALK activation in neuroblastoma as opposed to other malignancies (lung cancer) at the first opportunity.
3. Figure 1 is very informative (although the text in the legend requires formatting), and although the authors state that liquid biopsies have been reviewed extensively elsewhere, for the Open Biology readers it would be useful to give some key examples of clinical translation for the different types of circulating biomarkers in other malignancies.
4. Section 3.3 is very long. Subtitles here would be welcome.
5. "et al" is occasionally italicised, but mostly not.
6. Section 4 is a good translational summary and discussion which could be extended to provide further "constructive discussion and/or critique of the field" (as per Open Biology guidelines). The need for a final further summary (section 5) is questionable.

Author's Response to Decision Letter for (RSOB-19-0056.R0)

See Appendix A.

Decision letter (RSOB-19-0056.R1)

23-Apr-2019

Dear Dr Trigg

We are pleased to inform you that your manuscript entitled "Opportunities and challenges of circulating biomarkers in neuroblastoma" has been accepted by the Editor for publication in Open Biology.

Sincerely,

The Open Biology Team
mailto: openbiology@royalsociety.org

Appendix A

UNIVERSITY OF CAMBRIDGE
DEPARTMENT OF PATHOLOGY
DIVISION OF CELLULAR AND MOLECULAR PATHOLOGY

Level 3 Lab Block, Box 231,
Addenbrooke's Hospital, Hills Road, Cambridge, CB2 0QQ
Tel: 01223 336911 – Email: rt473@cam.ac.uk

Dr Ricky M. Trigg, PhD
Postdoctoral Research Associate

April 16, 2019

Dear Buchi,

We are delighted that our manuscript has been accepted for publication in *Open Biology* and would like to sincerely thank the reviewer for their time and very helpful feedback.

Please see below a point-by-point response to the reviewer's comments. Changes in the manuscript are shown in red.

1. The INSS staging system is so central to the discussion of each biomarker, that it deserves more explanation in the introduction. A table would be useful.

We have now provided Table 1 – a description of the INSS stages, and have also stated that the INSS system is post-surgical.

2. ALK (and ATRX) should be defined at the first mention in section 1.

We have now defined the biological roles of ALK and ATRX at the first mention.

For the readership of this journal it could be useful to compare and contrast the nature of ALK activation in neuroblastoma as opposed to other malignancies (lung cancer) at the first opportunity.

We have now contrasted the activation of full-length ALK in ALK+ NB with the fusion of ALK to other oncogenes in ALK+ ALCL and NSCLC.

3. Figure 1 is very informative (although the text in the legend requires formatting), and although the authors state that liquid biopsies have been reviewed extensively elsewhere, for the Open Biology readers it would be useful to give some key examples of clinical translation for the different types of circulating biomarkers in other malignancies.

We have now formatted the text in the figure legend. We have now introduced two examples of clinically translated liquid biopsy tests: the Cobas test for EGFR mutation testing in NSCLC and the CELLSEARCH test for enumeration of CTCs in several cancer types.

4. Section 3.3 is very long. Subtitles here would be welcome.

We have now split this section into three further sections: the first is an introduction to CTCs, the second is methodology of CTC detection and the third is association of CTCs with clinical features.

5. "et al" is occasionally italicised, but mostly not.

We have now italicised every instance of *et al*.

6. Section 4 is a good translational summary and discussion which could be extended to provide further "constructive discussion and/or critique of the field" (as per Open Biology guidelines). The need for a final further summary (section 5) is questionable.

We have added some more narrative to the discussion on clinical implementation of circulating biomarkers in NB. We feel that the Summary is concise and brings the review to an end; however, we will leave the decision of whether to include or exclude the Summary with the Editorial team.

We have also designed a graphical abstract to appear alongside our published article and feel it would also serve as a more informative Figure 1, so have substituted it.

Thank you for your time and we look forward to reading the proofs of our manuscript in due course.

Kind regards,

Ricky Trigg